# The Learning Transfer of Dementia Training Program Participants: Its Antecedents and Mediating Effect on the Job Competency of Geriatric Caregivers

**DOI:** 10.3390/healthcare11222991

**Published:** 2023-11-19

**Authors:** Chulwoo Kim, Jin Lee, Geon Lee

**Affiliations:** 1Department of Public Administration, Gachon University, Seongnam 13120, Republic of Korea; cwkim@gachon.ac.kr; 2School of Lifelong Education, Gachon University, Seongnam 13120, Republic of Korea; bsjldb@gachon.ac.kr; 3Department of Public Administration, Hanyang University, Seoul 04763, Republic of Korea

**Keywords:** geriatric caregiver, care worker, learning transfer, training transfer, job competency, dementia training program, nursing education, South Korea

## Abstract

This study examined how learning transfer and its antecedents impacted job competency among geriatric caregivers who underwent dementia training. The dementia training program of the National Health Insurance Service of South Korea was selected for this study. The participants included caregivers who provided care to patients with dementia at 3–12 months post-training. A hierarchical regression analysis was used to verify the research model. The results are summarized as follows: First, transfer opportunity and transfer intention were antecedent variables with a statistically significant positive effect on the near transfer of learning. Second, self-efficacy, the instructor’s role, and transfer intention were antecedent variables with a statistically significant positive effect on the far transfer of learning. Third, the near transfer of learning had a statistically significant effect on all six competency variables (communication, problem solving, interpersonal relationships, skills, self-development, and work ethics). Fourth, the far transfer of learning had a statistically significant static effect on all six competency variables, although the size of the influence on competency enhancement was relatively small compared with the near transfer of learning. This study confirmed the effects of various transfer climate-related variables in a training program on job competency, suggesting that the learning transfer of caregivers who underwent dementia training is a significant mediating variable. The limitations of this study and directions for future research are also discussed. The learning transfer of caregivers who underwent this training enhanced their job competencies.

## 1. Introduction

From 2015 to 2050, the percentage of the global population aged 60 and older is expected almost to double, increasing from 12% to 22%, and the number of individuals with dementia is projected to increase from 55 million in 2019 to reach 139 million by 2050 [1]. In South Korea, rapid population aging has resulted in a significant annual increase in the number of patients with dementia. Therefore, the South Korean government has included the “National Responsibility System for Dementia” in the “Five-Year Plan for State Administration”. Additionally, the expansion of 256 dementia care centers in the country has augmented the availability of professional caregivers for dementia.

Since July 2014, the National Health Insurance Service (NHIS) in South Korea has been conducting specialized dementia training for employees of long-term care institutions for grade 5 dementia. Specialized dementia training comprises courses for understanding dementia, care, facility care, home care, and program management [2].

Dementia requires different types of treatment and care depending on disease progression, symptoms caused by multiple cognitive impairments, and deterioration in daily life performance. This necessitates professional and systematic training as well as caregiver training for patients with dementia [3,4]. The systematized job training of caregivers for dementia can improve their knowledge of dementia and professional functions, helping to nurture specialized caregivers for dementia [5,6]. More importantly, the cultivation of professional knowledge and a positive attitude toward caring for people with dementia are the key requirements for a caregiver for older adults with dementia [7,8]. Dementia-related expertise secured through training can positively change caregiver attitudes toward older adults with dementia, which can be linked to improvements in the quality of long-term care services [9].

However, there are very few cases in which training is applied directly to the workplace. Previous studies have reported that most people do not utilize the knowledge and skills acquired during workplace training [10,11]. A perception survey on learning transfer targeting human resource development managers showed a 41% retention immediately after training, 24% six months after training, and 15% one year after training [12]. In terms of training investment, only 10% was transferred to work performance [13,14]. This implies that the training imparted in many programs is not applied in the workplace, which is contrary to pre-training expectations, and that most of the training costs are wasted because of low learning transfer, thereby inducing negative views regarding the necessity of human resource development training [15].

Therefore, it is important to ensure the effectiveness of specialized dementia training programs conducted by the NHIS. Learning transfer refers to the degree to which learning is applied from training to the workplace and is a critical factor for training effectiveness [16]. It is crucial to confirm the effectiveness of specialized dementia training programs in establishing a strategic plan for training transfer.

To provide better long-term care services, it is essential to augment caregivers’ work capacities. Caregiver competency is a key factor in determining the quality of service; therefore, an environment must be created to facilitate the acquisition of professional nursing skills [17]. A high level of learning transfer implies that trainees apply the knowledge, skills, and attitudes acquired through training in their actual work—that is, there is a high possibility of improving their work performance [18,19,20,21,22].

This study aims to provide an opportunity to increase the effectiveness of dementia training programs that are continuously expanding. It will also help to improve caregivers’ capabilities. A multidisciplinary professional service group requires skills to care for older people’s physical, mental, and social aspects as they enter an aging society.

However, no study has yet explored the effect of learning transfer, as well as its antecedents and their effects, on the job competency of geriatric caregivers. Therefore, this study aims to investigate the factors influencing the learning transfer of caregivers who received dementia training, and the mediating role of learning transfer on job competency. To achieve this goal, the following research questions are derived: (1) What is the relationship between the antecedents (trainee characteristics, training design, work environment, transfer intention) and learning transfer of caregivers who have undergone dementia training? (2) How does the learning transfer in these caregivers mediate the relationship between antecedents and job competency?

## 2. Theoretical Background

### 2.1. Learning Transfer

Kirkpatrick’s [23] four-level evaluation model of training effectiveness, the most widely used in the field of corporate training, provides a systematic perspective on the transfer of corporate training. Level 1 is a reaction evaluation, such as training satisfaction and instructor satisfaction; Level 2 is a learning evaluation, such as learning achievement; Level 3 is a behavior evaluation based on learning; and Level 4 is a result evaluation based on behavior. Learning transfer corresponds to Level 3 in the evaluation model—that is, behavior evaluation, which is defined as “evaluating the degree of behavioral change that occurred through training participants’ participation in training” [13,24,25,26,27].

Similarly, many scholars have considered learning transfer as the application of the knowledge, skills, and attitudes acquired during training in a work environment [12,28,29,30]. Learning transfer is classified into positive and negative transfers according to content [31], near and far transfers according to scope [32,33], and low- and high-road transfers according to level [34]. Based on the results of the studies mentioned above, learning transfer refers to a change in behavior and maintaining that change as a result of applying the knowledge, skills, and attitudes acquired through a workplace training program to a wider job area.

However, most studies do not explicitly define the concept of learning transfer. Primarily, the meaning of “near transfer”, that is, the application of the training content equally to the field, is utilized as the concept of “learning transfer”, and empirical research has attempted to identify its predictive variables [13,24]. However, empirical research on far transfer, that is, the application of training content to a wider range of situations, is lacking [35,36,37,38,39].

This lack of research could be due to the comparative rarity and difficulty of measuring far transfer [36,38,39]. However, compared to the past, the significance of far transfer, that is, the creative application of learning content to various situations, is rapidly increasing [40]. Clark [36] emphasizes that the ability to apply principles to various situations is a crucial requirement for knowledge workers to double organizational productivity.

The above definitions of transfer suggest that while all transfers have some common features in terms of applying what has been learned, there are differences in terms of the content, level, and scope of application. By combining the various definitions above, this study conducts additional discussions by defining near transfer as “applying the knowledge, skills, and attitudes, learned through dementia training, appropriately to the situation of the older people with dementia”, and far transfer as “creatively applying the knowledge, skills, and attitudes, learned through dementia training, to the care situation of the older people without disabilities”. In general, caregivers’ tasks for older adults with reduced physical function and those with dementia and reduced cognitive function contain the characteristics of both near and far transfers but in a complex arrangement. Moreover, the goal of a training program is to simultaneously pursue both near and far transfers.

The learning transfer measurement period can be as short as two weeks post-training and as long as 18 months [13,24]. Thus, learning transfer should be measured after trainees who have undergone training are provided with the opportunity to perform transfer behaviors [41]. Accordingly, recent learning transfer studies have measured learning transfers at least four weeks after learning.

Regarding the timing of measuring the learning transfer, there is no established standard regarding the appropriate time to measure it post-training, and there are various temporal differences depending on the focus of a study [42,43,44]. Some results differ depending on the measurement period [45,46].

In this study, data were collected from the caregivers of a facility dedicated to dementia and a cognitive activity-type visiting care institution in a metropolitan area. These caregivers were chosen from those belonging to long-term care institutions who passed the final evaluation after completing 60 h of the basic training course, visiting care course, or facility course among the dementia training courses conducted by the NHIS in South Korea at 3–12 months post-training [46,47,48,49].

### 2.2. Antecedents of Learning Transfer

The variables that have been studied and identified thus far that affect learning transfer can be broadly divided into three types: trainer characteristics, training design (training content, training method, trainer ability), and work environment factors (support from superiors and peers, transfer opportunities) [25,50]. The following types of variables can be applied based on caregiver characteristics:

First, the learning motivation and self-efficacy of nursing care workers who obtained national certification without restrictions on education or age were set as learner characteristic factors.

Second, the training content, teaching method, and trainer ability for specialized dementia training were set as training design factors for nursing care workers who had never received group training for an extended period other than nursing caregiver training.

Third, in the 1:1 service provision work environment between older people with dementia and their caregivers, the trainees applied influencing factors such as opportunities to use new knowledge or technology, support from superiors, and support from colleagues as work environment factors. Considering this specificity, it is possible to confirm the effect of transfer intention on caregivers’ professional dementia training.

Empirical studies have shown that learning motivation, transfer design, training satisfaction, and self-efficacy are determinants of learning transfer [51,52]. Additionally, training methods, content, and characteristics have been found to affect learning transfer [53].

In addition, the trainees’ attitudes can be an essential factor. To improve the transfer effect of training, a trainer’s willingness to transfer must be considered. For a successful training transfer to occur in the workplace, an environment must be provided in which the training content is applied systematically and in the same situation and manner. Therefore, trainees’ continued willingness to find and apply what they have learned in situations where they can apply it is an essential factor in learning transfer [54].

From the individual’s perspective, promoting the transfer of education and training is achieved through intention. Intention is a concept that explains an individual’s cognitive will to perform a specific action to elaborate on the structure of training transfer [55]. As such, transfer intention is essential for transfer, but empirical research on this variable still needs to be conducted [56]. The measurement of transfer intentions showed diverse results depending on the study. Furthermore, more empirical verification of the role of transfer intention in transferring training is required.

### 2.3. Competency

McClelland [57] defined “competency” as a broad range of psychological and behavioral characteristics related to work performance. McLagan [58] defined it as the characteristic of an individual’s outstanding performance in a job or role. Dubois [59] viewed competency as based on personal characteristics used to successfully play a role in life. Parry [60] found that it significantly affected aspects of an individual’s work and was highly related to work performance, regarding it as a collection of knowledge, skills, and attitudes that could be measured against performance standards widely accepted in an organization and improved through training and development.

Spencer and Spencer [61] define competency as “an individual’s internal characteristics leading to effective and superior performance according to criteria in a specific situation or job”, considering deeper aspects such as individual motives, traits, and self-concepts that lead to such behavior beyond the aspects of visible knowledge or skills. Abraham et al. [62] defined it as a complete set of characteristics, behaviors, and traits necessary for successful job performance. Combining all the definitions, Salman et al. [63] defined competencies as “the personality attributes or characteristics such as knowledge, skills, attitudes, abilities, motives, and self-concept demonstrated by employees that result in effective and superior performance”.

In South Korea’s long-term care insurance system, caregivers are defined as members of the workforce who provide physical and housekeeping support services for older adults who have difficulty independently performing daily life activities because of geriatric diseases such as dementia or stroke [64]. Few studies have explored the types of job roles they perform in the workplace and the required job competencies. Competency refers to the individual performance resulting from the application of what has been learned in the field. Thus, competency refers to measuring an individual’s performance.

To improve the professionalism of caregivers who specialize in care in nursing facilities, the professional competencies required in terms of the knowledge that care workers must possess are derived from health and medical knowledge regarding dementia and geriatric diseases, and an understanding of older adults and their families. Additionally, the professional competencies required in terms of skills are extracted, such as professional counseling, the identification of needs, personal care, and administrative practice skills. The professional competencies required in terms of value should be augmented in terms of emotional empathy with users and their families, the ability to induce emotional stability to relieve users’ anxiety, and a customer service mindset that regards users as welfare service customers [65,66,67].

In this study, performance resulting from transfer is classified as individual-level performance. The degree of improvement in competency due to learning transfer over a certain period is the difference between the level of competency possessed by the trainee before the implementation of the training program and that after a period of at least three weeks to one year has elapsed post-training. In this study, competency is defined as a personal outcome and an outcome variable of transfer.

The competency model identified during curriculum development includes definitions of competency and behavioral indicators required for job performance, where behavioral indicators are behaviors exhibited by excellent employees. Therefore, competency and behavioral indicators indicate the content of training and facilitate the evaluation of the training effects. Given this definition of competency, learning transfer in a competency-based curriculum can be interpreted as implying that trainees possess behavioral indicators of competency. As Lee [68] proved that the transfer of learning between the work environment and competency had a significant effect as a mediating variable, learning transfer was expected to have a mediating effect between trainee characteristics, training design, work environment, transfer intention factors, and competency, which is the result of learning transfer. Based on this, the following hypotheses are proposed:

**Hypothesis** **1.**
*The antecedent variables (trainee characteristics, training design, work environment, and transfer intention) positively affect learning transfer.*


**Hypothesis** **1a.**
*The antecedent variables positively affect the near transfer of dementia training for trainees.*


**Hypothesis** **1b.**
*The antecedent variables positively affect the far transfer of dementia training for trainees.*


**Hypothesis** **2.**
*Learning transfer among dementia training trainees positively affects job competency.*


**Hypothesis** **2a.**
*The near transfer of dementia training to trainees positively affects job competency.*


**Hypothesis** **2b.**
*The far transfer of dementia training for trainees positively affects job competency.*


**Hypothesis** **3.**
*Learning transfer has a mediating effect when the antecedent variables (trainee characteristics, training design, work environment, and transfer intention) affect job competency.*


**Hypothesis** **3a.**
*The near transfer of learning has a mediating effect.*


**Hypothesis** **3b.**
*The far transfer of learning has a mediating effect.*


## 3. Study Design

### 3.1. Operational Definitions of Variables

The antecedent variables used in this study are categorized into four domains: trainee characteristics, training design, work environment, and intention to transfer. The operational definitions of the sub-factors of the antecedent variables are presented in Table 1. Learning transfer, used as a mediating variable, is defined as a trainee’s perception of the degree of application of knowledge and skills acquired in the collective training process to the workplace.

In this study, competency was considered as the dependent variable and defined as the level of competency retention after a certain training period. In the context of this study, the relevant competencies included communication; problem solving; self-development; and interpersonal, technical, and work ethics competencies. Based on the National Competency Standards (NCS), among the 10 basic vocational competencies for augmenting the caregiver’s competency, six of the key job competencies required to perform the care competency unit were applied to derive core behaviors according to competency.

### 3.2. Subjects

#### 3.2.1. Criteria for Selecting Training Programs for Participants

In this study, the pattern of training transfer among the training participants was confirmed, with a focus on a specific dementia training program. Accordingly, two types of selection criteria were applied to select the study subjects. First, the criteria were considered because of the nature of the research related to training transfer. Therefore, we referred to the selection criteria presented in previous studies on training transfer [40,43,69,70]. Second, there were criteria limiting the attributes of the dementia training programs.

Selection criteria 1: Selection criteria based on the attributes of training transfer research

First, while all training provided by organizations is aimed at improving work or performance, when training more directly impacts business performance, the transfer of training becomes a key issue. In this study, the application of learning content in the field was selected as a significant training program at the organizational level.

Second, to ensure an adequate number of cases, we selected a training program in which as many trainees participated as possible. As ordinary training programs do not accommodate such a significant number of participants, programs with the same content and methods were selected for several rounds, and a program that could secure the maximum number of cases within realistic limits was selected.

2.Selection criteria 2: Dementia training program selection criteria

First, the dementia training program conducted by the NHIS encompasses everyone, including caregivers in long-term care institutions and institutional managers; however, the responsibility and professionalism of caregivers who determine service quality in organizations is increasing. Therefore, this study focused on a dementia training program for caregivers with the most practical implications for long-term care services.

Second, a dementia training program that operated in the form of collective training for a short period of less than one month was selected. According to Hughes, Ginnett, and Curphey [71], most job training programs in organizations last less than a week. This tendency appears more explicitly in training for middle managers than for top management, as middle managers play a key role in workplace workflow. Many job training programs for middle managers in South Korean companies comprise short-term courses for two to three days.

Third, the dementia training program comprehensively included interpersonal relationships, communication, goal management, performance management, and change management.

Fourth, in terms of training methods, programs that introduced various teaching methods, not just lecture-based training, were selected. Among the variables included in this study, the training design factors included teaching strategies to promote self-management and progress to motivate students. Considering that recent dementia training programs apply various methods, such as group assignments, discussions, and problem-solving teaching methods, this can be viewed as an appropriate selection criterion.

#### 3.2.2. Training Program for Participants: Dementia Specialized Training through Long-Term Care Insurance for Older Adults

The NHIS program was selected as the training program for this study. The dementia training program includes members of occupational groups belonging to various long-term care institutions, and its curriculum is designed to provide cognitive activity services for patients with dementia.

The caregiver course in the dementia training program comprised 40 h of the basic course and 20 h of the visiting or facility care course, for a total of 60 h of training. The trainees participated in the dementia training program’s core basic learning activities, beginning with an orientation to the curriculum on the first day and continuing until the fifth day, when activities to organize the learning content and prepare an action plan in the field were performed. During the eight-day training period, various teaching methods were used, including lecture-style classes, individual activities and assignments, individual competency self-assessments, individual practice plans, and group discussion assignments. Appendix A presents the curriculum of the dementia training program.

After the completion of the curriculum, a comprehensive evaluation was conducted. Only those who passed the evaluation were eligible to care for older adults with dementia. The survey focused on improving the curriculum and evaluating the overall satisfaction, difficulty level, composition, teaching methods, and instructors.

#### 3.2.3. Selection of Study Participants

Among the long-term care institutions in Korea, cognitive activity-type visiting and inpatient care facilities dedicated to patients with dementia in Seoul and Gyeonggi-do were randomly selected. The specific participant recruitment process was as follows: A survey was conducted targeting 400 caregivers who provided dementia care services in Seoul and Gyeonggi-do. As of 2017, there were 19,917 long-term care institutions nationwide, with 3002 in Seoul and 4704 in Gyeonggi-do, for a total of 7706 (38.7%) long-term care institutions in Korea. There were 326,417 active caregivers nationwide, with 61,392 in Seoul (18.8%) and 79,183 in Gyeonggi (24.3%), accounting for 43.1% of all caregivers in the country [72]. These areas were selected for this study because Seoul represents large cities, and Gyeonggi covers rural areas, small- and medium-sized cities, and large cities.

After explaining the purpose and method of the study to officials from long-term care institutions and obtaining their written consent, the contact information and work schedules of potential participants who had completed dementia training in 2016 and 2017, conducted through the NHIS, were obtained. This was intended to provide adequate opportunities for trainees who participated in the training to perform sufficient transfer behaviors after learning, and to allow flexibility in the timing of learning transfer.

In accordance with the work schedule of potential participants, the authors visited the institution, explained the purpose and method of the study, and obtained consent from interested participants. The survey was fixed for a day when a substantial number of participants could participate; those who were unable to participate because of their work schedules were excluded. The study method was re-explained at the survey location, and 400 individuals participated.

Ethical approval was obtained from the university committee affiliated with the first author. Ethical considerations related to research involving human participants, such as informed consent and confidentiality, were strictly observed. Engagement in the survey was entirely voluntary and respondents had the option to skip any questions they did not want to answer. Additionally, they were free to decline participation or withdraw from the study at any time. Responses were kept anonymous and confidential to ensure privacy.

### 3.3. Analytical Scales and Methods

#### 3.3.1. Time and Target of Measurement

Previous studies have frequently used three months as the time period to measure the time required for learning transfer; thus, the researchers intended to provide a sufficient period for learning transfer to occur, as open skills were the primary skills acquired in the dementia training program [46]. Leitl and Zempel-Dohmen [73], who explored transfer motivation as a concept similar to transfer intention as a variable attribute, measured transfer motivation immediately post-training and after three months to identify the variables that affected the change in transfer motivation. Based on the studies discussed above, the time of measurement after the completion of training ranged from a minimum of 3 months to a maximum of 12 months in this study.

#### 3.3.2. Composition of Measurement Scales

In this study, the levels of 17 variables in six areas (trainee characteristics, training design, work environment, transfer intention, learning transfer, and competency) were measured. The questionnaire for the nine sub-factors comprising trainee characteristics, training design, work environment, and intention to transfer, which are the antecedent variables of this study, was constructed from related studies in Korea and reconstructed to fit the dementia training environment based on the studies by Saks and Belcourt [48], Lee [74], and Park and Kim [75]. As for the work environment (supervisory support, colleague support, and transfer opportunity), Park and Kim [75] adapted and utilized scales from overseas research and reconstructed them to fit the work environment of the caregivers. The mediating variables (near and far transfers) were used to reconstruct the questionnaire items of Holton [76] and Keum and Chung [15], which were similar to caregivers’ work environments. Regarding competency, which was a dependent variable in the NHIS study [2] on the training of long-term care workers, the six job competencies most required by caregivers in the care field were constructed based on the National Competency Standards (NCS) data and reconstructed accordingly.

A preliminary survey was conducted to ensure the face validity of each measurement scale, and items that were duplicated or gave ambiguous expressions in the flow of all items were modified or deleted. For items highlighted in terms of clarity and comprehension, the vocabulary was modified and supplemented. Additionally, the scale for measuring the transfer of training was reviewed by the instructor for each module of dementia training based on the training goals and contents.

#### 3.3.3. Data Analysis and Statistical Processing

First, outliers were identified through scatter plots and standardized scores, in addition to frequency analysis, to identify missing values and incorrectly written items in the data and to identify the general characteristics of the participants. Second, the mean, standard deviation, skewness, and kurtosis were analyzed using descriptive statistics to determine whether the main variables had a normal distribution. Third, an exploratory factor analysis was conducted to establish the sub-factors of each scale used in this study. Fourth, a confirmatory factor analysis was conducted to prove the validity of the sub-factors derived from the exploratory factor analysis. Fifth, a reliability analysis was conducted to ascertain the internal consistency of the sub-factors revealed through the exploratory factor analysis. Sixth, a Pearson’s moment time correlation analysis was conducted to determine whether there was a positive or negative correlation between the main variables. Seventh, a hierarchical multiple regression analysis was conducted to identify the relationship between the independent and dependent variables, and the mediating effect was verified using the Sobel test. AMOS 23.0 and SPSS 23.0 for Windows were used for the statistical analysis, and the confidence level in all statistical analyses was set at 95%, with a significance level of 0.05.

As the data were gathered through self-reported questionnaires from a single source, there might have been concerns regarding the potential for common method bias. To address this issue, we conducted Harman’s single-factor test on the primary variables in our research model, which included the dependent, independent, and mediating variables. The single factor explained 30.26% of the variance, which is significantly lower than the 50% threshold. These results indicate that the study did not suffer from substantial common method bias.

## 4. Results

### 4.1. Study Variable General Statistic

#### 4.1.1. Descriptive Statistics of Research Variables

A total of 400 items were distributed in this survey, of which 279 copies (valid data rate: 69.8%) were used for analysis, excluding 11 incomplete responses out of 290 copies (recovery rate: 72.5%). Regarding the characteristics of the respondents, women accounted for 84.9%, the age group of 50–59 years accounted for 43.0%, high school graduates accounted for 40.1%, those with no dementia training experience prior to the dementia training provided by the NHIS accounted for 64.5%, and those who were at least 12 months from the completion of dementia training accounted for 17.9% of the participants. Participants with at least three years of experience accounted for 48.4%, those without a family member with dementia accounted for 55.9%, those with visiting care as their work type accounted for 71.3%, and those with no certification other than a caregiver license, even after checking whether they had overlapping licenses, accounted for 44.6%.

Seventeen variables were measured in this study, and Table 2 presents the descriptive statistics. For each variable, the average value of the items selected through the validation process was used, and the measured values ranged from a minimum of one to a maximum of five. This showed that the average of each variable representing the normality of distribution was 3.49–4.70, with the trainer ability (3.49) having the lowest level. The training method (3.72), instructor role (3.49), supervisor support (3.83), and peer support (3.77) had a mean of four or less, with a standard deviation greater than 0.42. Univariate normality was assessed using skewness and kurtosis indices among the descriptive statistics. As the skewness of all variables used in this study revealed a distribution between −1.545 and 0.008, and the kurtosis displayed a distribution between −0.787 and 2.332, normality was secured, enabling statistical analysis.

#### 4.1.2. Factor Analysis and Reliability Analysis

The Cronbach’s alpha values of all variables were high at 0.80 or higher, except for self-efficacy. In particular, the reliability values of variables such as the instructor’s role, training method, supervisor support, peer support, work ethics, and intention to transfer were above 0.90. In conclusion, the reliability of the scale used in this study is very high. As a result of the factor analysis of the three items of transfer intention, it was extracted as one factor. The factor loadings for transitional intention ranged from 0.812 to 0.964, the eigenvalue for transitional intention was 2.339 (over 1.0), and the total explained variance was 77.98%, indicating that the factor was valid.

As a result of the factor analysis of the nine items of learning transfer, far transfer (four items) and near transfer (five items) were extracted as sub-factors. The factor loadings for far transfer ranged from 0.628 to 0.953 and those for near transfer ranged from 0.758 to 0.847. As the eigenvalues of the two factors for learning transfer were greater than 1.0 and the explanatory power for the total variance was 70.39%, the items for learning transfer were found to have factor validity. The results of analyzing the reliability of nine items for learning transfer revealed a very high internal consistency of 0.8 or higher in all five items for near transfer (Cronbach’s α = 0.898) and four items for far transfer (Cronbach’s α = 0.930).

According to the results of the factor analysis obtained using the maximum likelihood and direct oblimin methods, interpersonal relationships, work ethics, communication, skills, problem solving, and self-development factors were extracted as sub-factors of the concept of competency. The factor loadings for interpersonal relationships ranged from 0.522 to 0.973, those for work ethics from 0.641 to 0.858, those for communication from 0.764 to 0.824, those for technology from −0.803 to −0.902, those for problem solving from −0.688 to −0.825, and those for self-development from 0.547 to 0.582. The eigenvalues of all six factors for these competencies were above 1.0, and the explanatory power of the total variance was 69.01%, indicating factor validity. Additionally, the results of the reliability analysis of the 24 items on competency showed that all had a high internal consistency of 0.8 or higher.

#### 4.1.3. Measurement Model

##### Verification Results and Goodness of Fit Analysis Results for the Measurement Model

Table 3 presents the results of examining the fit index of the measurement model used in the confirmatory factor analysis of this study. In other words, the goodness of fit of the measurement model consisting of nine antecedent variables, such as learning motivation, two mediating variables, and one dependent variable, was 2110.316 (df = 1014, *p* < 0.001), with an RMR of 0.029 and an RMSEA of 0.062, indicating a good level. For a TLI of 0.891 and a CFI of 0.902, the TLI was slightly below 0.9, but not significantly low. The PNFI was slightly higher than the 0.6 threshold at 0.744. Overall, this was considered to be applicable to the analysis.

##### Parameter Estimation Results for the Measurement Model

According to the results of the confirmatory factor analysis, the observed factor loadings for nine latent antecedent variables, two latent mediating variables, and one latent dependent variable are shown in the tables presented below. Table 4 presents the unstandardized and standardized coefficients of the factor loadings of the nine latent variables corresponding to the antecedent variables.

According to the table above, a statistical significance of *p* < 0.001 was observed for all nine latent variables, including learning motivation. All standardized factor loadings were 0.7 or higher, and the CR values were higher than 1.965, confirming the appropriate constructs for each latent variable. Additionally, as another indicator of the convergent validity of the latent variables, the degree of the average variance extraction (AVE) and construct reliability were identified. In all the nine latent variables, the conceptual reliability indicated a value of 0.7 or more, and the AVE revealed a high value of 0.5 or more, indicating convergent validity. The latent variables extracted from each index variable had appropriate conceptual reliability.

Table 5 presents the estimated factor load for the latent variable competency corresponding to the dependent variable, and the results confirming convergent validity. All standardized factor loadings converged to a high value of 0.5 or more, the conceptual reliability for this converged to a high value of 0.7 or more, and the AVE revealed a high value of 0.5 or more, confirming that the measurement variables for the latent variables were appropriately converged.

Finally, Table 6 presents the estimated results of the near and far learning transfers, which are the mediating variables in the research model. According to the results, the standardized factor loadings for the two latent variables all converged to a value of 0.5 or higher, the concept reliability was 0.7 or higher, and the AVE was higher than 0.5, confirming that the latent variables extracted from each measurement variable were appropriately converged.

##### Results of Discriminant Validity Verification of Measurement Model

The results of the correlations between the latent variables are as follows: Among the training designs, the trainer ability and training method (r = 0.709, *p* < 0.01) had the strongest positive correlations. In the work environment, peer and supervisor support (r = 0.612, *p* < 0.01) revealed statistically significant positive correlations with the interpersonal relationships and self-developmental competencies (r = 0.668, *p* < 0.01). The mediating variables, the near transfer of learning (r = 0.522, *p* < 0.01) and far transfer of learning (r = 0.497, *p* < 0.01), displayed the highest positive correlations with transfer intention. There was a positive correlation between the dependent variable of problem-solving competency and the mediating variable of the near transfer of learning (r = 0.540, *p* < 0.01). Communication competency and the antecedent variable of transfer intention (r = 0.417, *p* < 0.01) showed a positive correlation. In terms of correlation, through the correlation analysis between the variables used in this research model, the correlation coefficients of all variables were found to be 0.80 or less, and there were no multicollinearity issues.

However, as another indicator to confirm the discriminant validity of these 17 latent variables, a comparison between the square value of the correlation coefficient and the AVE value of the latent variable shows that the AVE value was greater than the squared value of the correlation coefficient, securing discriminant validity for all latent variables.

### 4.2. Hypothesis Verification Results

#### 4.2.1. Relationship between Antecedent Variables and Learning Transfer (Near Transfer/Far Transfer)

Table 7 presents the results of the multiple regression analysis of the near transfer of learning. This regression model was confirmed to be suitable for analysis, as indicated by the F-value, which indicated the goodness of fit of the model and was statistically significant at 10.586 (*p* < 0.001). It had an R^2^ of 0.408 and an explanatory power of 40.8% for the total variance. The D/W was close to 2 at 1.642, indicating that the assumption of independence of the residual was satisfied. Additionally, multiple regression analyses can be suspicious of multicollinearity because of the high correlation between variables. The analysis suggested no problem regarding multicollinearity as the tolerance limit was 0.1 or less, and the variance inflation coefficient (VIF) was between 1 and 10.

A verification of the significance of the antecedent variables based on the results of the multiple regression analysis of the near transfer of learning revealed that, among the input variables, self-efficacy (β = 0.131, t = −2.032, *p* < 0.05), transition opportunity (β = 0.300, t = 4.775, *p* < 0.001), and transition intention (β = 0.275, t = 4.152, *p* < 0.001) had a statistically significant effect on the near transfer of learning. When demographic variables were controlled, the near transfer of learning increased by 0.276 (B = 0.276) with an increase in the transfer opportunity of one unit, whereas the near transfer of learning increased by 0.265 (B = 0.265) with an increase in the transfer intention of one unit. This indicates that the influences of transfer intention and transfer opportunity, which correspond to the characteristics of the work environment, were extremely similar in increasing the near transfer of learning.

Next, a multiple regression analysis of the far transfer of learning was conducted to investigate the influence of trainees’ individual characteristics, training design, and work environment characteristics. Table 8 presents the results of this analysis. The goodness of fit of the regression model for the far transfer of learning was found to be statistically significant, with F at 10.728 (*p* < 0.001) and R^2^ at 411, explaining 41.1% of the total variance and making it suitable for analysis. Additionally, the D/W (1.716) satisfied the assumption of the independence of residuals, confirming that there was no problem with the multicollinearity test statistics through tolerance limits and VIF.

An examination of the statistically significant influencing factors based on the results of the multiple regression analysis on the far transfer of learning showed that β was −0.129 for the work experience variable (t = −2.413, *p* < 0.05) and β was 0.140 for the monthly income variable (t = 2.001, *p* < 0.05). Among the antecedent variables, β was 0.152 for the training method (t = 2.080, *p* < 0.05), 0.168 for the trainer ability (t = 2.172, *p* < 0.05), and 0.370 for transfer intention (t = 5.609, *p* < 0.001), which was observed to have statistical significance, indicating that the influence of transfer intention was the greatest. In other words, for every unit increase in transfer intention, the far transfer of learning was estimated to increase by 0.394.

#### 4.2.2. Verification Results for the Mediating Effect of Learning Transfer

In this study, the mediating variables were divided into near and far learning transfers, as presented in the research model. A hierarchical regression analysis was used to verify the mediating effect, which was divided into two categories. Additionally, the competency variable corresponding to the dependent variable was divided into six sub-factors: communication, problem solving, self-development, interpersonal relationships, skills, and work ethics competencies; the mediating effect for each of these was verified. This mediating effect was reviewed based on the premise of the one-level model analyzed above, in which the antecedent variable had a statistically significant effect on the mediating variable. Regarding the mediating effect of the near transfer of learning, the variables of self-efficacy, transfer opportunity, and transfer intention were analyzed, and the mediating effects of the far transfer of learning, training method, instructor role, and transfer intention were analyzed. Hypotheses 1 and 2 were verified based on the results of the hierarchical regression analysis, and a Sobel test was conducted to verify each mediating effect. Appendix A present the results for the mediating effects of near and far learning transfer.

#### 4.2.3. Verification of the Significance of Mediating the Effect of Learning Transfer

Table 9 tabulates the results of the Sobel test to confirm the mediating effect of the near transfer of learning as well as the statistical significance. Multiple regression and hierarchical regression analyses confirmed the mediating effect of the near transfer of learning as well as its statistical significance. According to the results of the Sobel test, among the mediating effects of the near transfer of learning derived through the hierarchical regression analysis, transfer intention had a partial mediating effect on all six competency variables. The statistical significance of the mediating effect was verified at a significance level of 0.05.

Furthermore, among the antecedent variables with statistical significance confirmed in the one-level model, in the case of the transfer opportunity, the complete mediating effect of the near transfer of learning was confirmed in the effect on problem-solving competency, in addition to confirming statistical significance in the Sobel test. Regarding the relationship with the other five competency variables, there was no mediating effect of the near transfer of learning. Finally, regarding self-efficacy, a partial mediating effect of the near transfer of learning was confirmed in the relationship between communication competency and problem-solving competency. Additionally, the full mediating effect of the near transfer of learning was confirmed for the interpersonal, skills and self-development competencies, whereas no mediating effect was confirmed for work ethics competency. According to the Sobel test results, no statistical significance was observed for any of the six competency variables. Finally, the mediating effect of the near transfer of learning had a statistically significant partial mediating effect only on the relationship between transfer intention and the competency variables.

On the one hand, the Sobel test results for the mediating effect of the far transfer of learning indicated that statistical significance was not secured in many influence relationships compared to the mediating effect of the near transfer of learning. This suggests that the strength of the mediating effect was unstable. Among the variables of the training method, trainer ability, and transfer intention, which were statistically significant in the first-stage model of hierarchical regression analysis, only the intention to transfer was consistently confirmed to have a partial mediating effect on all six competency variables, and the results of the Sobel test were statistically significant.

On the other hand, there was no mediating effect of the far transfer of learning in any of the six competency variables for the training method variables, and there was no statistical significance in the Sobel test. Regarding the trainer ability, only the partial mediating effect of the far transfer of learning was found to be statistically significant in the relationship between influence and communication competency, and the complete mediating effect of the far transfer of learning was confirmed in the relationship between problem-solving competency and self-development competency. The Sobel test showed no statistical significance, with no mediating effect on skills or work ethic competencies. Table 10 presents the results of the significance test on the mediating effect of the far transfer of learning.

A hierarchical regression analysis, which confirmed the mediating effects of the near and far transfers of learning on the relationship between the influence of nine antecedent and six competency variables, showed that the mediating variables, the near and far transfers of learning, had a partial mediating effect on the relationship between transfer intention and the influence of competency variables. There was a lack of consistency or statistical significance in the relationship between the influence of other antecedent and competency variables.

## 5. Discussion

### 5.1. Antecedent Variables and Learning Transfer

Among the antecedent variables, self-efficacy, transfer intention, and transfer opportunity were shown to have a direct effect on the near transfer of learning. This is consistent with the results of existing studies, wherein it has been proven that the higher the self-efficacy, the more likely a successful completion of the curriculum and transfer of learning content to the field [13,24,77]. Next, the influence of transfer opportunities corresponding to the work environment factor is relatively large. This may be because the caregivers who participated in the training were more exposed to situations in which they had to apply their learning to the care work site immediately after training.

Therefore, a dementia training program should be tailored for immediate workplace application, necessitating the establishment of an integrated system that links training with practical field experiences [78]. Such an approach is essential for enhancing transfer intention, facilitating practical learning transfer, and providing continuous opportunities for near transfer. Notably, enhancements in this domain are imperative, as the current training design demonstrates a limited impact on the concept of the near transfer of learning.

In continuously maintaining the far transfer of learning as an index to confirm the continuity of the dementia training effect, the trainee’s individual intention to transfer, as well as the training method and trainer ability during training design, have a significant impact. Among these, the intention to transfer was found to have the strongest influence. In other words, the results suggest that the level of transfer intention of the individual trainee is more significant than that of any other variable in ensuring that the learning results are utilized in the field while maintaining continuity.

### 5.2. Mediating Effect of Learning Transfer

This study examines the mediating effect of learning transfer on various competencies—communication, problem solving, self-development, interpersonal relations, technical skills, and professional ethics—among trainees engaged in a dementia training program. The results suggest that the level of transfer intention of individual trainees was the most decisive factor for participants undergoing dementia training to apply their learning in the field and improve their competency [79].

Furthermore, this study indicates that far learning transfer may have a relatively lesser impact when compared to near learning transfer and that its influence may be limited for certain variables. Therefore, future studies should explore strategies to augment the efficacy of far learning transfer. The aforementioned outcomes furnish critical insights for the formulation of educational strategies aimed at enhancing learner competencies, offering foundational data not only for the amelioration of dementia education but also for the improvement of pedagogical practices in diverse learning environments.

## 6. Conclusions

### 6.1. Implications of the Study

The following implications are suggested based on this study’s results: First, the intention to transfer significantly impacts learning transfer and competency and requires a systematic approach. This may be because trainees who undergo dementia training are more exposed to caring for older adults with dementia in their workplaces after training. Therefore, the possibility of conversion to near or far learning transfers varies according to the level of metastatic intention. Dementia training programs should be designed for immediate workplace applications. In addition, an operating system linked to training and the field should be used to facilitate the near transfer of learning, which is a practical training effect. Furthermore, continuous management through systematic and regular practical training is required to facilitate the application of acquired knowledge and skills in the field.

Second, the dementia training method and trainer ability appear to impact the far transfer of learning, suggesting the need to formulate a plan for them. In the case of training dealing with open skills, such as dementia training, the cramming method of teaching to deliver theoretical knowledge via one-time group training should be considered with caution. Therefore, dementia training program instructors should comprehend the characteristics of the workplace and caregivers of long-term care institutions, emphasize trainees’ active participation, examine case management presentations, and follow established practices regarding the application of learning in the field [80]. Additionally, it is necessary to prepare experience-sharing mechanisms, such as providing time for dementia training participants to discuss issues related to situations and coping experiences in the workplace.

Third, caregivers who have completed dementia training must actively participate in training and programs to enhance their self-efficacy, create a supportive work environment for transition opportunities, and make active organizational efforts to increase transition intentions. These findings imply that trainees with high self-efficacy and self-regulation, as well as more utilization opportunities in their workplace, apply their acquired knowledge and skills more frequently. The key factor is the organization’s support, for example, whether members’ performance is appropriately evaluated and whether the organization considers the improvement of job competency through training or material support for members. In other words, the near transfer of learning suggests that the individual characteristics of trainees, the far transfer of learning, and the characteristics of the training system are relatively significant influencing factors. The level of the trainee’s transfer intention indicates that both near and far learning transfers are continuous influencing factors.

Fourth, the learning transfer of caregivers who underwent dementia training enhanced their job competencies. However, it is necessary to develop specific measures to enable people caring for patients with dementia in the workplace to gain consciousness as dementia care professionals. However, the significance of competencies such as the ability to induce emotional empathy and emotional stability in older adults with dementia and a sincere service mindset should not be overlooked, in addition to the role of assisting the planned program in the workplace. Efforts should be made to provide training opportunities and develop content that can improve these aspects.

### 6.2. Limitations of the Study

This study is meaningful because it verifies the effect of dementia training on the job competency of caregivers who underwent dementia training through learning transfer and suggests practical and policy alternatives to help caregivers continuously demonstrate competency in the workplace. Nevertheless, this study has a few limitations, which warrants follow-up research.

First, the data collection process was limited. In this study, 279 caregivers who participated in dementia training in Seoul and Gyeonggi-do were randomly sampled, limiting the generalizability of the results. Second, in terms of the research methodology, in addition to the self-report method (single-source/mono method), it is necessary to objectively examine the relationship between learning transfer and competency through case studies or qualitative research methods. These include in-depth interviews and additional comparative analyses of trainees’ manager data. This will facilitate an investigation into the meaning of learning transfer for caregivers specializing in dementia, work environment by institution, and care work experience.

## Figures and Tables

**Table 1 healthcare-11-02991-t001:** Operational definitions of variables.

Variable Type	Classification	Sub-Factor	Definition
Antecedent	Trainee Characteristic	Learning motivation	Tendency and willingness to make the effort to apply the knowledge and skills acquired through training in the workplace.
		Self-efficacy	A general personal belief that one can change one’s job performance if desired.
	Training Design	Training content	The degree to which the trainee judges whether the training content reflects the requirements of the actual job.
		Training method	The training methods that the instructor employs to effectively convey the contents of training to the participants and enhance their understanding.
		Trainer ability	The degree to which the instructor effectively communicates the training content to facilitate its understanding and the formulation of a plan for use in the field by the trainees.
	Work Environment	Supervisor support	The trainee’s perception of the extent to which the supervisor reinforces their ability to apply their training in the workplace.
		Peer support	The degree to which peers reinforce and support when trainees apply new learning to work.
		Transfer Opportunity	The degree to which resources or tasks are provided to enable the trainees to apply new learning from training to work.
	Transfer Intention	Transfer intention	The level of willingness of trainees to apply the contents newly learned during training.
Mediating	Learning Transfer	Near transfer	New learning from training is applied to specific and limited tasks.
		Far transfer	New learning from training is generally and widely applied in various situations.
Dependent	Competency	Communication	The ability to accurately understand what others mean and to precisely convey one’s intentions orally and in writing.
		Problem solving	The ability to identify and solve problems during job performance creatively and logically.
		Self-development	The ability to manage and develop oneself in the pursuit of one’s duties.
		Interpersonal relationship	The ability to get along well with people who interact with the caregiver during the performance of work without causing problems.
		Skills	The ability to apply appropriate tools and devices when performing work.
		Work ethic	The attitudes, manners, and occupational perspectives required during job performance.

**Table 2 healthcare-11-02991-t002:** Descriptive statistics of variables (*n* = 279).

Measurement Variable	Minimum	Maximum	Mean	Standard Deviation	Skewness	Kurtosis
Learning motivation	3.00	5.00	4.69	0.43	−1.270	0.856
Self-efficacy	3.00	5.00	4.40	0.50	−0.256	−0.787
Training content	2.00	5.00	4.17	0.59	−0.273	−0.118
Training method	1.50	5.00	3.72	0.78	−0.566	0.106
Trainer ability	1.50	5.00	3.49	0.86	−0.108	−0.637
Supervisor support	1.75	5.00	3.83	0.86	−0.441	−0.610
Peer support	1.50	5.00	3.77	0.73	−0.207	−0.051
Transfer opportunity	2.67	5.00	4.44	0.56	−0.735	−0.330
Transfer intention	3.00	5.00	4.50	0.54	−0.783	−0.064
Near transfer	2.80	5.00	4.33	0.52	−0.312	−0.515
Far transfer	3.00	5.00	4.19	0.57	−0.117	−0.517
Communication	2.50	5.00	4.33	0.53	−0.404	−0.230
Problem solving	2.33	5.00	4.23	0.58	−0.382	−0.025
Skills	2.67	5.00	4.20	0.54	0.008	−0.449
Self-development	3.00	5.00	4.43	0.53	−0.803	−0.109
Work ethics	2.67	5.00	4.70	0.42	−1.545	2.332
Interpersonal relationships	2.00	5.00	4.26	0.56	−0.415	0.126

**Table 3 healthcare-11-02991-t003:** Results of goodness of fit analysis for the measurement model.

Index	χ^2^ (df = 1014)	RMR	RMSEA	PNFI	CFI	TLI
Model	2110.32 *p* < 0.001	0.029	0.062	0.744	0.902	0.891

**Table 4 healthcare-11-02991-t004:** Results of factor loading estimation based on the antecedent variables (*n* = 279).

Measurement Variable	Latent Variable	Factor Loading		S.E.	C.R.	AVE	Reliability
		Unstandardized	Standardized				
Learning motivation 1	Learning Motivation	1	0.892			0.906	0.974
Learning motivation 2		1.013	0.924	0.044	22.786 ***		
Learning motivation 3		1.025	0.853	0.052	19.683 ***		
Learning motivation 4		0.64	0.631	0.053	11.986 ***		
Self-efficacy 1	Self- Efficacy	1	0.87			0.794	0.918
Self-efficacy 2		1.003	0.866	0.062	16.128 ***		
Self-efficacy 3		0.754	0.549	0.08	9.404 ***		
Training content 1	Training Content	1	0.847			0.805	0.942
Training content 2		0.991	0.849	0.057	17.281 ***		
Training content 3		0.993	0.879	0.055	18.149 ***		
Training content 4		0.747	0.666	0.061	12.204 ***		
Training method 1	Training Method	1	0.764			0.810	0.944
Training method 2		1.153	0.874	0.073	15.873 ***		
Training method 3		1.295	0.897	0.079	16.392 ***		
Training method 4		1.464	0.93	0.086	17.105 ***		
Trainer ability 1	Role of Instructor	1	0.856			0.757	0.925
Trainer ability 2		1.067	0.888	0.054	19.794 ***		
Trainer ability 3		1.037	0.906	0.051	20.523 ***		
Trainer ability 4		0.948	0.792	0.058	16.272 ***		
Supervisor support 1	Manager Support	1	0.86			0.811	0.945
Supervisor support 2		1.131	0.892	0.055	20.484 ***		
Supervisor support 3		0.932	0.861	0.049	19.178 ***		
Supervisor support 4		1.095	0.938	0.048	22.592 ***		
Peer support 1	Peer Support	1	0.895			0.813	0.945
Peer support 2		1.098	0.946	0.043	25.629 ***		
Peer support 3		1.018	0.9	0.044	22.925 ***		
Peer support 4		0.917	0.717	0.062	14.754 ***		
Transfer opportunity 1	Transfer Opportunity	1	0.818			0.811	0.928
Transfer opportunity 2		0.974	0.765	0.073	13.26 ***		
Transfer opportunity 3		1.183	0.822	0.083	14.26 ***		
Transfer intention 1	Transfer Intention	1	0.831			0.913	0.969
Transfer intention 2		1.107	0.939	0.055	19.947 ***		
Transfer intention 3		1.082	0.881	0.059	18.411 ***		

*** *p* < 0.001.

**Table 5 healthcare-11-02991-t005:** Results of factor loading estimation for dependent variables (*n* = 279).

Measurement Variable	Latent Variable	Factor Loading		S.E.	C.R.	AVE	Reliability
		Unstandardized	Standardized				
Self-development	Competency	1	0.75	0.75		0.788	0.957
Skills		0.962	0.711	0.083	11.634 ***		
Problem solving		1.089	0.758	0.088	12.44 ***		
Communication		0.941	0.716	0.08	11.719 ***		
Work ethics		0.57	0.539	0.066	8.677 ***		
Interpersonal relationships		1.064	0.765	0.085	12.562 ***		

*** *p* < 0.001.

**Table 6 healthcare-11-02991-t006:** Results of analysis for mediating variables (*n* = 279).

Measurement Variable	Latent Variable	Factor Loading		S.E.	C.R.	AVE	Reliability
		Unstandardized	Standardized				
Learning transfer 3	Near transfer	1	0.857			0.823	0.959
Learning transfer 4		0.887	0.818	0.053	16.793 ***		
Learning transfer 5		0.845	0.746	0.058	14.559 ***		
Learning transfer 2		0.87	0.797	0.054	16.116 ***		
Learning transfer 1		0.889	0.773	0.058	15.363 ***		
Learning transfer 8	Far transfer	1	0.931			0.898	0.972
Learning transfer 9		0.992	0.907	0.039	25.76 ***		
Learning transfer 7		0.958	0.909	0.037	25.898 ***		
Learning transfer 6		0.821	0.771	0.047	17.533 ***		

*** *p* < 0.001.

**Table 7 healthcare-11-02991-t007:** Results of multiple regression analysis on near transfer of learning (*n* = 279).

Model		Unstandardized Coefficient		Standardized Coefficient	t	Significance Probability	Collinearity Statistic	
		B	Standard Error				Tolerance	VIF
	(constant)	0.964	0.403		2.394	0.017		
Control variable	Sex (Male_dum)	−0.170	0.088	−0.117	−1.948	0.053	0.626	1.598
	Age	0.004	0.030	0.008	0.143	0.886	0.810	1.235
	Training level	0.023	0.031	0.043	0.761	0.447	0.713	1.402
	Work experience	−0.013	0.030	−0.024	−0.448	0.655	0.787	1.271
	Training experience	0.000	0.028	0.000	−0.002	0.998	0.832	1.202
	Monthly income	0.029	0.028	0.072	1.028	0.305	0.458	2.181
	Training point	0.002	0.008	0.013	0.259	0.796	0.894	1.118
	Work type (Facility_dum)	0.112	0.085	0.098	1.322	0.187	0.415	2.411
Antecedent Variable	Learning motivation	−0.046	0.072	−0.038	−0.648	0.517	0.645	1.551
	Self-efficacy	0.136	0.067	0.131	2.032	0.043	0.547	1.829
	Training content	0.017	0.059	0.019	0.283	0.777	0.497	2.012
	Training method	0.067	0.048	0.101	1.382	0.168	0.424	2.358
	Trainer ability	0.061	0.047	0.101	1.296	0.196	0.376	2.658
	Supervisor support	−0.065	0.045	−0.107	−1.431	0.154	0.404	2.474
	Peer support	0.054	0.046	0.076	1.174	0.241	0.537	1.861
	Transfer opportunity	0.276	0.058	0.300	4.775	0.000	0.575	1.739
	Transfer intention	0.265	0.064	0.275	4.152	0.000	0.518	1.930
F = 10.586 ***, R^2^ = 0.408 (Adjusted, 0.370), Durbin–Watson = 1.642								

*** *p* < 0.001.

**Table 8 healthcare-11-02991-t008:** Results of multiple regression analysis on far transfer of learning (*n* = 279).

Model		Unstandardized Coefficient		Standardized Coefficient	t	Significance Probability	Collinearity Statistic	
		B	Standard Error				Tolerance	VIF
	(constant)	0.444	0.444		1.000	0.318		
Control variable	Sex (Male_dum)	−0.150	0.096	−0.093	−1.556	0.121	0.626	1.598
	Age	0.004	0.033	0.006	0.120	0.905	0.810	1.235
	Training level	0.024	0.034	0.040	0.707	0.480	0.713	1.402
	Work experience	−0.079	0.033	−0.129	−2.413	0.017	0.787	1.271
	Training experience	0.055	0.031	0.092	1.763	0.079	0.832	1.202
	Monthly income	0.061	0.031	0.140	2.001	0.046	0.458	2.181
	Training point	−0.016	0.008	−0.094	−1.863	0.064	0.894	1.118
	Work type (Facility_dum)	0.152	0.094	0.119	1.620	0.107	0.415	2.411
Antecedent variable	Learning motivation	0.031	0.079	0.023	0.387	0.699	0.645	1.551
	Self-efficacy	0.143	0.074	0.125	1.938	0.054	0.547	1.829
	Training content	−0.066	0.065	−0.069	−1.027	0.305	0.497	2.012
	Training method	0.111	0.053	0.152	2.080	0.039	0.424	2.358
	Role of instructor	0.112	0.052	0.168	2.172	0.031	0.376	2.658
	Manager support	−0.057	0.050	−0.086	−1.150	0.251	0.404	2.474
	Peer support	0.095	0.051	0.121	1.863	0.064	0.537	1.861
	Transfer opportunity	0.124	0.064	0.122	1.945	0.053	0.575	1.739
	Transfer intention	0.394	0.070	0.370	5.609	0.000	0.518	1.930
F = 10.728 ***, R^2^ = 0.411 (Adjusted, 0.373), Durbin–Watson = 1.716								

*** *p* < 0.001.

**Table 9 healthcare-11-02991-t009:** Result of testing the significance of the mediating effect of near transfer of learning.

Dependent Variable	Antecedent Variable (Level 1)	Unstandardized Coefficient			Sobel Statistic		Mediation Type
		Level 2	Level 3	Mediating Effect Size	z Value	*p*	
Communication competency	Self-efficacy	0.190	0.137	0.053	1.928	0.054	Partial
	Transfer opportunity	0.040	−0.067	0.107	3.769	0.0001	-
	Transfer intention	0.312	0.209	0.103	3.439	0.0005	Partial
Problem-solving competency	Self-efficacy	0.287	0.229	0.058	1.927	0.0539	Partial
	Transfer opportunity	0.171	0.053	0.118	3.759	0.0002	Full
	Transfer intention	0.240	0.126	0.114	3.431	0.0006	Full
Interpersonal relationship competency	Self-efficacy	0.148	0.108	0.04	1.848	0.0646	Full
	Transfer opportunity	0.044	−0.038	0.082	3.258	0.0012	-
	Transfer intention	0.307	0.229	0.078	3.038	0.0024	Partial
Skills competency	Self-efficacy	0.182	0.136	0.046	1.879	0.0602	Full
	Transfer opportunity	0.033	−0.060	0.093	3.437	0.0005	-
	Transfer intention	0.268	0.179	0.089	3.181	0.0013	Partial
Self-development competency	Self-efficacy	0.152	0.109	0.043	1.884	0.0596	Full
	Transfer opportunity	0.110	0.022	0.088	3.468	0.0005	-
	Transfer intention	0.275	0.190	0.085	3.243	0.0012	Partial
Work ethics competency	Self-efficacy	−0.079	−0.114	0.035	1.857	0.0633	-
	Transfer opportunity	0.102	0.032	0.07	3.307	0.0009	-
	Transfer intention	0.197	0.130	0.067	3.078	0.0021	Partial

**Table 10 healthcare-11-02991-t010:** Result of testing the significance of the mediating effect of far transfer of learning.

Dependent Variable	Antecedent Variable (Level 1)	Unstandardized Coefficient			Sobel Statistic		Mediation Type
		Level 2	Level 3	Mediating Effect Size	z Value	*p*	
Communication competency	Training method	−0.030	−0.064	0.034	1.803	0.0714	-
	Trainer ability	0.175	0.140	0.035	1.999	0.0455	Partial
	Transfer intention	0.312	0.189	0.123	3.889	0.0001	Partial
Problem-solving competency	Training method	−0.070	−0.087	0.017	1.519	0.1286	-
	Trainer ability	0.128	0.111	0.017	1.542	0.1231	Full
	Transfer intention	0.240	0.181	0.059	2.056	0.0398	Partial
Interpersonal relationship competency	Training method	−0.016	−0.036	0.02	1.692	0.0907	-
	Trainer ability	−0.003	−0.023	0.02	1.725	0.0849	-
	Transfer intention	0.307	0.238	0.069	2.556	0.0106	Partial
Skills competency	Training method	0.077	0.047	0.03	1.89	0.0588	-
	Trainer ability	0.051	0.021	0.03	1.933	0.0532	-
	Transfer intention	0.268	0.161	0.107	3.46	0.0005	Partial
Self-development competency	Training method	−0.128	−0.150	0.022	1.782	0.0747	-
	Trainer ability	0.032	0.009	0.023	1.818	0.0691	Full
	Transfer intention	0.275	0.196	0.079	2.938	0.0037	Partial
Work ethics competency	Learning motivation	0.013	−0.001	0.014	1.581	0.1138	-
	Peer support	−0.049	−0.063	0.014	1.607	0.1082	-
	Transfer intention	0.197	0.148	0.049	2.217	0.0266	Partial

## Data Availability

The data presented in this study are available on request from the corresponding author. The data are not publicly available due to the participants of this study not providing written consent for their data to be shared publicly.

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
