# Peer review of "The Learning Transfer of Dementia Training Program Participants: Its Antecedents and Mediating Effect on the Job Competency of Geriatric Caregivers"

_healthcare, 2023, doi:10.3390/healthcare11222991_

Round 1
Reviewer 1 Report
Comments and Suggestions for Authors
This is a comprehensive report of an extensive and careful study about the learning transfer in a training program for dementia caregivers.
I have no major concern about the manuscript in its version 2.
Minor point: I suggest to write at the end of the Abstract (as in Conclusion): The learning transfer of the caregivers who underwent this training enhanced their job competencies.
Author Response
Thank you for your comment and suggestion. At the abstract section, I put the sentence, "The learning transfer of caregivers who underwent this training enhanced their job competencies."
Reviewer 2 Report
Comments and Suggestions for Authors
1 - The introduction can be improved, given that up to bibliographical reference 13, references between the years 1992 and 2011 are included.
2 - "aims to investigate the factors influencing the learning transfer of caregivers who received dementia training, and the relationship between learning transfer and competency." (p.2). However, the second sub-question asks for a «yes» or «no» (2) Does learning transfer play a mediating role in the relationship between these antecedent variables and job competency?
3- The theoretical background, in the competence part, authors and references from the 70s to 90s are used, without mentioning the more recent concept of competence, and the combination of knowledge, skills and attitudes.
4 - The data was collected from participants in training in the years 2016 and 2017. Is there any difference in the subjects of the 2016 studies or 2017?
5 - Results and Hypothesis Verification Results are clear and strong. However, the conclusions are quite synthetic and there is no discussion of other findings/results. The discussion of the results could be expanded and/or deepened.
6 - the most recent bibliographic reference is one from 2019, followed by two from 2017 and 2015. The form of bibliographic review is omitted - nevertheless, the presence of the subject "Learning Transfer of the Dementia Training Program" in the databases is considerable.
Author Response
We would like to express our appreciation to the peer reviewer who provided thoughtful and scholarly evaluations of our manuscript. We tried to apply all your suggestions. We marked the revised part of the manuscript in green.
Response to the Reviewer 2
1 - The introduction can be improved, given that up to bibliographical reference 13, references between the years 1992 and 2011 are included.
Response: The articles we referenced were the key articles in training research that other scholars also referenced in their recent publications. Additionally, we referenced the following four articles published in the last five years.
Brion, C. (2020). Learning Transfer: The Missing Linkage to Effective Professional Development. Journal of Cases in Educational Leadership, 23(3), 32–47. https://doi.org/10.1177/1555458920919473
Gkioka, M., Schneider, J., Kruse, A., Tsolaki, M., Moraitou, D., & Teichmann, B. (2020). Evaluation and Effectiveness of Dementia Staff Training Programs in General Hospital Settings: A Narrative Synthesis with Holton's Three-Level Model Applied. Journal of Alzheimer's Disease, 78(3), 1089–1108. https://doi.org/10.3233/JAD-200741
Weiss, J., Tumosa, N., Perweiler, E., Forciea, M. A., Miles, T., Blackwell, E., Tebb, S., Bailey, D., Trudeau, S. A., & Worstell, M. (2020). Critical Workforce Gaps in Dementia Education and Training. Journal of the American Geriatrics Society, 68(3), 625–629. https://doi.org/10.1111/jgs.16341
Yun, J., Kim, D., & Park, Y. (2019). The influence of informal learning and learning transfer on nurses' clinical performance: A descriptive cross-sectional study. Nurse Education Today, 80, 85–90. https://doi.org/10.1016/j.nedt.2019.05.027
2 - "aims to investigate the factors influencing the learning transfer of caregivers who received dementia training, and the relationship between learning transfer and competency." (p.2). However, the second sub-question asks for a «yes» or «no» (2) Does learning transfer play a mediating role in the relationship between these antecedent variables and job competency?
Response: To clarify the research purpose, we edited the sentence you quoted and changed the research questions as follows.
(1) What is the relationship between antecedents (trainee characteristics, training design, work environment, transfer intention) and learning transfer of caregivers who have undergone dementia training? (2) How does the learning transfer in these caregivers mediate the relationship between antecedents and job competency?
3- The theoretical background, in the competence part, authors and references from the 70s to 90s are used, without mentioning the more recent concept of competence, and the combination of knowledge, skills and attitudes.
Response: Additionally, we referenced the following three articles on competency published in the last five years.
Jeyathevan, G., Cameron, J. I., Craven, B. C., & Jaglal, S. B. (2019). Identifying Required Skills to Enhance Family Caregiver Competency in Caring for Individuals With Spinal Cord Injury Living in the Community. Topics in Spinal Cord Injury Rehabilitation, 25(4), 290–302. https://doi.org/10.1310/sci2504-290
Salman, M., Ganie, S. A., & Saleem, I. (2020). Employee Competencies as Predictors of Organizational Performance: A Study of Public and Private Sector Banks. Management and Labour Studies, 45(4), 416–432. https://doi.org/10.1177/0258042X20939014
Song, Y., Chun, D., Xiong, P., & Wang, X. (2022). Construction of Talent Competency Model for Senior Care Professionals in Intelligent Institutions. Healthcare, 10, 914. https://doi.org/10.3390/healthcare10050914
4 - The data was collected from participants in training in the years 2016 and 2017. Is there any difference in the subjects of the 2016 studies or 2017?
Response: As mentioned in the manuscript on page 3,
In this study, data were collected from caregivers of a facility dedicated to dementia and a cognitive activity-type visiting care institution in a metropolitan area. These care-givers were chosen from those belonging to long-term care institutions who passed the final evaluation after completing 60 hours of the basic training course, visiting care course, or facility course among the dementia training courses conducted by the NHIS in South Korea at 3–12 months post-training.
The participants were chosen in 2016 and 2017 due to the post-training period, which we used as a research subject selection criterion.
5 - Results and Hypothesis Verification Results are clear and strong. However, the conclusions are quite synthetic and there is no discussion of other findings/results. The discussion of the results could be expanded and/or deepened.
Response: Although some discussions were dispersed through the result section. We combined them and expanded them in the newly added discussion section on page 18-19.
6 - the most recent bibliographic reference is one from 2019, followed by two from 2017 and 2015. The form of bibliographic review is omitted - nevertheless, the presence of the subject "Learning Transfer of the Dementia Training Program" in the databases is considerable.
Response: We added 12 recent bibliographic reference including five articles below.
Fauth, F., & González-Martínez, J. (2021). On the Concept of Learning Transfer for Continuous and Online Training: A Literature Review. Education Sciences, 11(133). https://doi.org/10.3390/educsci11030133
Fauth, F., & González-Martínez, J. (2022). Trainees’ Personal Characteristics in the Learning Transfer Process of Permanent Online ICT Teacher Training. Sustainability, 14(386). https://doi.org/10.3390/su14010386
Kim, D. (2021). Development and Effect of Virtual Reality Practice Program for Improving Practical Competency of Caregivers Specializing in Dementia. Healthcare, 9(10), 1390. https://doi.org/10.3390/healthcare9101390
Kim, E.-J., Park, S. and Kang, H.-S.(T). (2019), "Support, training readiness and learning motivation in determining intention to transfer", European Journal of Training and Development, Vol. 43 No. 3/4, pp. 306-321. https://doi.org/10.1108/EJTD-08-2018-0075
Kim, S.-O. (2021). Effect of Case-Based Small-Group Learning on Care Workers’ Emergency Coping Abilities. International Journal of Environmental Research and Public Health, 18(21), 11458. https://doi.org/10.3390/ijerph182111458
Reviewer 3 Report
Comments and Suggestions for Authors
The manuscript highlights learning transfer via a statistical model based on survey responses. The manuscript is appropriate for publishing pending minor changes as discussed below.
Language
The language is appropriate, with minor bits-and-bobs picked up while reading. These do not impact the readability, and thus during review it would be of benefit to double-check on sentence structures.
Introduction
The introduction highlights the problem appropriately and discusses the conceptual framework well.
Methods
Include the statistical models for normality and reliability testing at this stage already to orientate the reader. From a generalisation perspective, what generalisability is aimed for within the population? Although this is remarked during the limitations, allowing for discussion of the margin of error or delineation of the study results earlier on frames the research nicely for discussion.
Results and discussion
Although presented well, the statistical elements becomes quite heavy-handed. I recommend that the sections be summarised a tad more to allow for more emphasis of the impact of the results and their potential use for making inferences from the data. Supporting this with literature furthermore provides imperative for additional assessments, such as the qualitative measures discussed in the conclusion.
Author Response
We would like to express our appreciation to the peer reviewer who provided thoughtful and scholarly evaluations of our manuscript. We tried to apply all your suggestions. We marked the revised part of the manuscript in green.
Response to the Reviewer 3
The manuscript highlights learning transfer via a statistical model based on survey responses. The manuscript is appropriate for publishing pending minor changes as discussed below.
Language
The language is appropriate, with minor bits-and-bobs picked up while reading. These do not impact the readability, and thus during review it would be of benefit to double-check on sentence structures.
Response: We received additional editing service after this revision
Introduction
The introduction highlights the problem appropriately and discusses the conceptual framework well.
Response: Thank you
Methods
Include the statistical models for normality and reliability testing at this stage already to orientate the reader. From a generalisation perspective, what generalisability is aimed for within the population? Although this is remarked during the limitations, allowing for discussion of the margin of error or delineation of the study results earlier on frames the research nicely for discussion.
Response: The total population of South Korea stands at 51 million, with the inhabitants residing in the Seoul and Gyeonggi regions amounting to 26 million. Consequently, if a survey were to be conducted focusing on the metropolitan area, it could be posited that it represents approximately 50% of the entire populace, thus ensuring a certain degree of representativeness for the caregiving professionals. Concurrently, it can be assessed that the representativeness for caregivers operating in non-urban, suburban, or rural regions may be considered inadequate.
Results and discussion
Although presented well, the statistical elements becomes quite heavy-handed. I recommend that the sections be summarised a tad more to allow for more emphasis of the impact of the results and their potential use for making inferences from the data. Supporting this with literature furthermore provides imperative for additional assessments, such as the qualitative measures discussed in the conclusion.
Response: It is an extensive study so we tried to summarized and manage the overall lengths of the section into ten tables. On page 18-19, we newly added discussion section to explain the result deeply.